# Convolutional neural network using magnetic resonance brain imaging to predict outcome from tuberculosis meningitis

Trinh Huu Khanh Dong[1,2]*, Liane S. Canas[2], Joseph Donovan[1,3], Daniel Beasley[2], Nguyen Thuy Thuong-Thuong[1,4], Nguyen Hoan Phu[1,6], Nguyen Thi Ha[1,5], Sebastien Ourselin[2], Reza Razavi[2], Guy E. Thwaites[1,4], Marc Modat[2]

**1** Oxford University Clinical Research Unit, Viet Nam, **2** King's College London, United Kingdom, **3** London School of Hygiene and Tropical Medicine, London, United Kingdom, **4** Centre for Tropical Medicine and Global Health, Nuffield Department of Medicine, University of Oxford, Oxford, United Kingdom, **5** the Hospital for Tropical Diseases, Ho Chi Minh City, Viet Nam, **6** School of Medicine, Vietnam National University – Ho Chi Minh City, Viet Nam

* trinhdhk@outlook.com.vn

**Data availability statement:** Data cannot be shared publicly because of various reasons. To make requests for data sharing, the researchers

## Abstract

### Introduction

*Tuberculous meningitis* (TBM) leads to high mortality, especially amongst individuals with HIV. Predicting the incidence of disease-related complications is challenging, for which purpose the value of brain magnetic resonance imaging (MRI) has not been well investigated. We used a convolutional neural network (CNN) to explore the complementary contribution of brain MRI to the conventional prognostic determinants.

### Methods

We pooled data from two randomised control trials of HIV-positive and HIV-negative adults with clinical TBM in Vietnam to predict the occurrence of death or new neurological complications in the first two months after the subject's first MRI session. We developed and compared three models: a logistic regression with clinical, demographic and laboratory data as reference, a CNN that utilised only T1-weighted MRI volumes, and a model that fused all available information. All models were fine-tuned using two repetitions of 5-fold cross-validation. The final evaluation was based on a random 70/30 training/test split, stratified by the outcome and HIV status. Based on the selected model, we explored the interpretability maps derived from the models.

### Results

215 patients were included, with an event prevalence of 22.3%. On the test set our non-imaging model had higher AUC (71.2% ± 1.1%) than the imaging-only model (67.3% ± 2.6%). The fused model was superior to both, with an average AUC = 77.3% ± 4.0% in the test set. The non-imaging variables were more informative in the HIV-positive group, while the imaging features were more predictive in the HIV-negative group. All three models performed better in the HIV-negative cohort. The interpretability maps show

shall kindly contact Dr. Joseph Donovan (joseph.donovan@lshtm.ac.uk) or Prof. Guy Thwaites (gthwaites@oucru.org) for enquiries. For formal application, the researchers should detail their request on a request form as detailed on the OUCRU's data-sharing website (https://www.oucru.org/data-sharing-policy/) and send to CTU@oucru.org.

**Funding:** This project received funding from the Wellcome Trust, the Wellcome/EPSRC Centre for Medical Engineering (203148/Z/16/Z), the AI Centre for Value-Based Healthcare, the Vietnam ICU Translation Applications Laboratory (VITAL) consortium (217650/Z/19/Z, 215010/Z/18/Z, 225167/Z/22/Z) and Oxford University Clinical Research Unit. All processing was run on the EPSRCsupported (EP/T022205/1) Joint Academic Data Science Endeavour (JADE) HPC cluster. The funders had no role in study design, data collection and analysis, decision to publish, or preparation of the manuscript.

**Competing interests:** The authors have declared that no competing interests exist.

the model's focus on the lateral fissures, the corpus callosum, the midbrain, and periventricular tissues.

### Conclusion

Imaging information can provide added value to predict unwanted outcomes of TBM. However, to confirm this finding, a larger dataset is needed.

## Introduction

*Tuberculous meningitis* (TBM) is the most devastating form of tuberculosis, a disease caused by the dissemination of *Mycobacterium tuberculosis* (*Mtb*) into the central nervous system. It represents 2-5% of all cases of tuberculosis but kills 30%-50% of patients despite the best available anti-tuberculosis chemotherapy, with poorer outcomes especially associated with HIV coinfection [1–3]. Those who survive are often left with long-term neurological disability.

Most complications and deaths occur within two months of diagnosis and starting anti-tuberculosis treatment [1,4,5]. Early identification of individuals at high risk of complications and death may facilitate more targeted monitoring and management. However, despite attempts, state-of-the-art predictive models have generally yielded areas under the curve (AUCs) of 76-77% [1,6–10].

Many clinical and laboratory variables have been linked to poorer outcomes from TBM. Two of the strongest predictors are the Glasgow coma score (GCS), which assesses the degree of coma, and the British Medical Research Council (MRC) grade, which is based on the GCS and the presence of focal neurological deficit [11]. Other known predictors of poor outcomes are HIV coinfection with low CD4+ cell count and infection caused by multidrug-resistant bacteria [12]. Recent studies also suggest that host genotypic variation (e.g. in the Leukotriene A4 Hydrolase gene) may also influence intracerebral inflammation response and thus prognosis [10,13].

Brain imaging has been used for decades in diagnosing and managing TBM [14–16]. Its importance in clinical management has been widely accepted, with magnetic resonance imaging (MRI) being generally superior to computer tomography (CT) due to its greater ability to visualise small lesions, particularly in the brain stem and posterior fossa [17].

The ability of brain MRI to predict neurological complications and death is less well defined. The spectrum of neuro-radiological abnormalities caused by TBM varies across individuals, including hydrocephalus, contrast-enhancement of the basal meninges and cisterns, infarcts, and tuberculomas (Fig 1) [12]. These abnormalities are associated with more severe disease. For example, in a study from 2020 [15], 94% of 48 patients with MRC grade II and above had abnormalities on brain MRIs. In less severe cases, this proportion is reduced to 85% [17]. Two major limitations of MRI are its high cost and the dependence on manual, expert reading, which is slow and results in variability in interpretation, especially in low-resource settings. These low dimensional readings might also not be capable of capturing the whole spectrum of information available in an MRI volume and thus overlook minor but relevant characteristics.

Our study aims to evaluate the added predictive value of brain MRI information to predict the occurrence of any neurological complication within 60 days from the first imaging session. To our knowledge, this is the first attempt to employ imaging information to predict TBM outcomes automatically from baseline information data only.

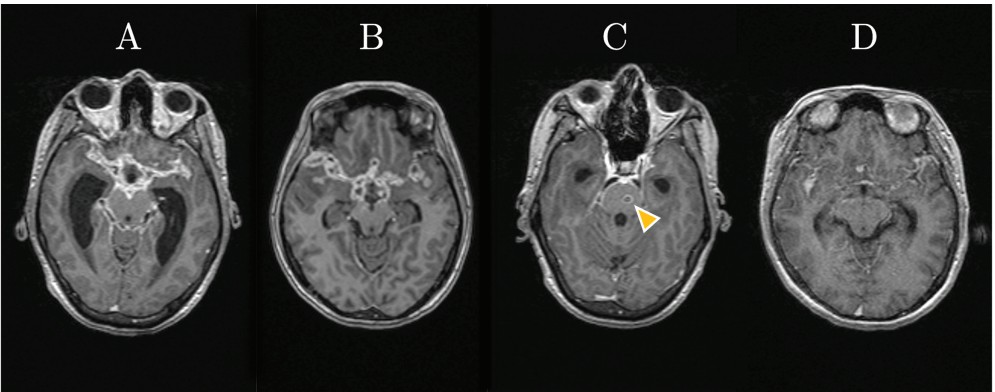

**Fig 1. Typical findings on MRI - T1 MPRAGE: A: basal enhancement, vasculitis with thickened vascular walls, and hydrocephalus with dilated lateral ventricles; B: basal enhancement; C: tuberculoma (arrow); D: vasculitis**

## Materials and methods

### Dataset

**Participants**   Participants were recruited from two randomised control trials (RCTs, identifiers NCT03092817 and NCT03100786) enrolling adults with TBM (HIV-positive or -negative) admitted to the Hospital for Tropical Diseases (HTD) in Ho Chi Minh City, Vietnam, from 25th May 2017 to 29th April 2021 [18–21]. Trial participants were treated with standard anti-tuberculosis drug therapy for 12 months, alongside a blinded 6-to-8-week tapering course of dexamethasone or placebo started at randomisation (with the exception of HIV-negative LTA4H TT-genotype participants in the LAST ACT trial who received open-label dexamethasone). All participants, or their relatives if they were incapacitated, provided written informed consent to enter the trials. The trials received ethical approvals from the Oxford Tropical Research Ethics Committee (approval numbers 36-16 and 52-16), the Vietnam Ministry of Health (approval numbers 38/CN-BDGDD, 108/CN-BDGDD, and 151/CN-BDGDD), and the HTD's local ethics committee (approval numbers 14/HDDD and 37/HDDD) [18].

Trial participants were recruited into this study if they were at least 18 years old at the time of enrolment; with a clinical diagnosis of TBM, i.e., at least five days of meningitis symptoms, and cerebrospinal fluid (CSF) abnormalities; with anti-tuberculosis chemotherapy already started or planned by the treating clinician. Exclusion criteria included: an additional (non-TBM) brain infection confirmed or suspected by any applicable tests; more than six consecutive days of two or more drugs active against *Mtb* or more than three consecutive days of any type of orally or intravenously administered corticosteroid immediately before randomisation; dexamethasone considered mandatory or contra-indicated for any reason; or if the patient had previously been randomised into this trial for a prior episode of TBM; and lack of consent [18–20]. In this analysis, we only included participants enrolled up until 2021 and had at least one usable T1-weighted (T1w) MPRAGE MRI volume within the first month from enrolment.

**Data collection**   Data was frozen and curated for the analysis on 1st June 2021. Demographics (age, gender and HIV status) and relevant medical history were collected at baseline. Routinely during follow-up, a panel of relevant prognostic variables were recorded, including MRC grade, GCS and CSF biomarkers. The date of any new neurological event and death

was documented (Table 1) [22]. Lumbar puncture was performed according to the respective study protocols, from which cerebrospinal fluid white cell counts and differentials, protein, lactate, and glucose (with paired blood glucose) were acquired. TBM diagnosis was subsequently determined according to published criteria [23].

Brain MRIs were taken by either a Siemens MAGNETOM ESSENZA or AVANTO in one designated hospital in Ho Chi Minh City. Participants were images as near to the start of anti-tuberculosis chemotherapy as possible and within one month from the start of anti-tuberculosis chemotherapy unless contra-indicated. In this analysis, three-dimensional (3D) T1-weighted brain MRI scans were acquired with magnetisation-prepared rapid acquisition (T1w—MPRAGE) pulse sequence. The scan's field of view was $168 \times 256 \times 160$ voxels, with spatial resolution = 1.00 mm $\times$ 1.00 mm $\times$ 1.15 mm. We quality checked (QC) all imaging data visually for their field of view coverage and any acquisition artefacts. All MR sessions that did not pass the QC were excluded. If there were multiple T1w-MRPRAGE scans acquired and passed the QC, the scan taken closest to the enrolment date was selected into the study, ensuring one scan per participant. A patient was excluded if no T1w-MPRAGE scan was available or passed the manual QC.

Prior to the study, identifiable information, including participant identification number, age, date/time, and laboratory biomarkers were pseudonymised by value shifting and variable standardisation. The relevant metadata of every MR scan was also removed and inaccessible.

**Feature encoding**  Amongst all non-imaging features, continuous variables were centred (demeaned) and scaled by twice the empirical standard deviation [24]. Those with right-skewed empirical distributions were log-transformed before the scaling. Binary variables were one-hot encoded (i.e. 0 for Negative/No and 1 for Positive/Yes). Missing variables were imputed within the training process using Multivariate Imputation by Chained Equations (MICE) using predictive mean matching [25], implemented in the R statistical package *mice* [25,26] and Python's *scikit-learn* [27].

All QC-passed MRI scans were first affine-registered into the 1mm isotropic MNI-template space [28] using NiftyReg package [29]. Registered volume dimensions were $180 \times 210 \times 180$ voxels$^3$. Spatially aligned scans were subsequently centre-cropped or -padded to $200 \times 200 \times 200$ voxels$^3$ in size. Image intensities were normalised to range [0, 1] to provide a relatively equivalent range to non-imaging features [24]. Registered and normalised images were visually checked before inputted into the analysis.

## Experiments

**Objective and primary endpoint**  The main objective of our study is to investigate the potential added values of imaging information in predicting the primary endpoint in TBM using early information from diagnostic time, complementing the state-of-the-art model which only uses conventional biomarkers [1].

In this study, the primary outcome was defined as any new occurrence of neurological complications, including death, within 60 days of treatment, using information from the baseline. The outcome definition followed the primary endpoints defined in the two RCTs

**Table 1. Modified MRC Grading for TBM.**

| Grade/Stage of TBM | Characteristics |
| --- | --- |
| I | GCS = 15, no focal neurology |
| II | GCS = 11-14 or GCS = 15 with any focal neurology |
| III | GCS < 11 |

[18–21]. Accordingly, a non-mortal neurological event was defined as a reduction of GCS at least two units for at least 48 hours or any new focal neurological deficit that occurred within the first two months of treatment [18,20]. We developed different models to investigate how well this outcome was predicted by conventional clinical assessments and laboratory biomarkers (forming the non-imaging model, $M_{clin}$), exclusively a brain MRI (which forms the imaging-only model, $M_{img}$), and finally, the fusion of both sources of information (fused model, $M_{fused}$).

**Non-imaging model**  We built a logistic regression model ($M_{clin}$) as a reference. Training for this reference model was done using the statistical package R, version 4.2.2 [26]. To enable a comparison with the literature, we only included features selected by a prior study, shared across HIV and non-HIV cohorts (Table 2 [1]), briefly: age, weight, HIV status, GCS, MRC grade, CSF lymphocyte count, CSF glucose (with corresponding paired blood glucose), CSF protein, and CSF lactate. To promote generalisation, we added a Ridge regularisation term $\lambda$ to the loss function of the model (i.e., Ridge regression, Eq 1).

Hence, we have

$$\hat{y} = \text{expit}(X^T w)$$

$$CE(y, \hat{y}) = -\sum_{i=1}^{n} y_i \log(\hat{y}_i) + (1 - y_i) \log(1 - \hat{y}_i) \qquad (1)$$

$$\mathcal{L} = CE(y, \hat{y}) + \lambda \|w\|_2,$$

where $X^T$ is a vector of features whose coefficients were represented by vector $w$; $X^T w$ is termed the linear predictor; $y = \{y_i\}$ and $\hat{y} = \{\hat{y}_i\}$ ($1 \leq i \leq n$) are the observed binary outcome vector of length $n$ and its corresponding predicted probability; $\lambda$ is the Ridge regularisation term. $CE(y, \hat{y})$ is the cross-entropy loss and $\mathcal{L}$ is the total loss function that is optimised.

**Incorporating imaging information**  To embed imaging information into the prognostic model, we developed a neural network consisting of three components. First, MRIs were processed using an *imaging features extractor* that consisted of ten convolutional blocks and followed by one adaptive average pooling and one fully connected (FC) layer. Then, a separated

**Table 2. Baseline characteristics, by HIV status.**

| Characteristic | N | Overall N = 215 | HIV-negative N = 136 | HIV-positive N = 79 | p-value |
|---|---|---|---|---|---|
| Event | 215 | 48 (22%) | 20 (15%) | 28 (35%) | <0.001 |
| Age (years) | 215 | 32 (24, 42) | 34 (27, 48) | 30 (24, 36) | 0.002 |
| Weight (kg) | 215 | 51 (47, 59) | 53 (48, 60) | 49 (45, 55) | 0.006 |
| Local neurological deficit | 212 | 29 (14%) | 16 (12%) | 13 (16%) | 0.4 |
| (Missing) | | 3 | 3 | 0 | |
| GCS | 213 | 15 (14, 15) | 15 (13, 15) | 15 (14, 15) | >0.9 |
| (Missing) | | 2 | 2 | 0 | |
| MRC Grade | 215 | | | | 0.6 |
| I | | 99 (46%) | 62 (46%) | 37 (47%) | |
| II | | 106 (49%) | 69 (51%) | 37 (47%) | |
| III | | 10 (4.7%) | 5 (3.7%) | 5 (6.3%) | |
| CSF lymphocyte count (cells/$\mu$L) | 207 | 77 (50, 87) | 81 (57, 87) | 71 (32, 86) | 0.019 |
| (Missing) | | 8 | 6 | 2 | |
| Median (1st, 3rd quartiles) for numeric variables; n (%) for categorical variables. | | | | | |
| Comparison was done by Wilcoxon rank sum test for continuous variables and Pearson's Chi square test for categorical variables | | | | | |
| Event is defined as any neurological event or death within 60 days of follow-up | | | | | |

shallow FC network was used to learn from the clinical and laboratory variables (*non-imaging features extractor*). Lastly, the *imaging features extractor* and *non-imaging features extractor* were fused into a three-FC-layer block, defined as the *classifier*. The overall architecture is shown in Fig 2.

Each block in the network consisted of one convolutional or linear layer, one drop-out layer, one normalisation layer, and one activation layer with Sigmoid Linear Unit (SiLU [30]) as the activation function, in that order. Apart from the first convolutional block of the *imaging features extractor*, every downsampling convolutional block (with stride = 2) was followed directly by an equi-resolution convolutional block (stride = 1). We used Instance Normalisation for the first three convolutional blocks and Batch Normalisation for the rest. 512 channels outputted from the *imaging features extractor* were averaged and flattened into a vector of 512 features before being compressed into a latent vector of length 24 using a fully-connected layer. No drop-out layer was used at the output block. The output layer has 2 dimensions corresponding to the two outcome classes.

We first experimented with an imaging-only model with only this latent vector fed into the *classifier*, forming $M_{img}$. This allowed us to explore the predictive value of imaging data alone in the context of missing other features. We finally incorporated clinical and laboratory features by concatenating the learnt information to the extracted imaging features (forming a latent vector of length 31) before feeding it into the *classifier* ($M_{fused}$).

Similarly to the non-imaging model, we used cross-entropy loss (Eq 1) with Sigmoid (expit) as the output activation function, denoted as $\mathcal{L}_{fused}$. We, however, did not implement the Ridge penalty $\lambda$ into the model. In its stead, a weight decay penalisation was used. To

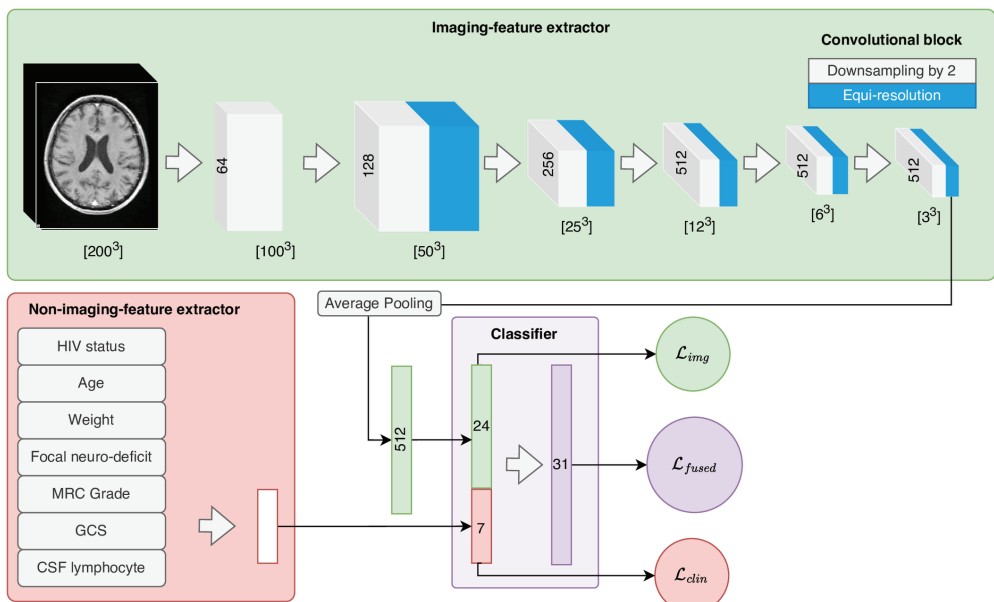

**Fig 2. Proposed Fused CNN architecture.** The network comprises of two branches: Imaging feature extractor (1, green-top) and Non-imaging feature extractor (2, red-bottom). The output of (1) is a 512x[3x3x3] array, which was subsequently average pooled into a 512x1 vector. This vector is compressed into a 24x1 vector using a fully-connected layer before being concatenated with the 7x1 vector outputted from branch (2). The 31x1 concatenated vector is fed into a Classifier (3, violet-bottom), which performs the classification tasks. During the training, classification was performed on both branch-specific latent vectors (forming $\mathcal{L}_{img}$ and $\mathcal{L}_{clin}$) and from the Classifier itself (forming $\mathcal{L}_{fused}$). However, in validation, only prediction from the Classifier was taken into account.

facilitate the extraction of relevant information and ensure equivalent attention to both imaging and non-imaging data, we also performed this classification task directly from the *imaging features extractor* and *non-imaging features extractor*. These auxiliary tasks were only used for training and were not considered during inference. The overall loss used for training was the weighted sum of the main loss $\mathcal{L}_{fused}$ and the two auxiliary losses ($\mathcal{L}_{img}$ and $\mathcal{L}_{clin}$) (Eq 2). For simplicity, in our study, we fixed $\alpha = \beta = 1$.

$$\mathcal{L} = \mathcal{L}_{fused} + \alpha \mathcal{L}_{img} + \beta \mathcal{L}_{clin} \tag{2}$$

**Model tuning, validation, and inferences of the selected model**  All models were optimised using the Joint Academic Data Science Endeavour (JADE) II cluster, with NVIDIA Tesla V100 32 GB GPUs. Before training, the total dataset was split into 30% for final assessment (herein denoted as test set) and 70% for training and validation (denoted as training set). The splitting was stratified by the response variable and HIV status. We used Novo-Grad [31] as the training optimiser.

Optimiser configurations (learning rate (LR) initial value and LR scheduler) and regularisation terms ($\lambda$ for $M_{clin}$ or dropout probability and weight decay for $M_{img}$ and $M_{fused}$) were tuned by a grid search utilising two repetitions of 5-fold cross-validation on the training set (S1 Fig). We selected the best model based on the average AUC and cross-entropy loss estimated on a pooled validation set. This pooled validation set was created by stacking the prediction and ground truth from ten left-out parts. The definition for each hyper-parameter is given in S1 Table.

At the first step of training, MRI scans were randomly augmented by (1) random flipping, (2) random affine transformation (shearing, scaling, rotation, and translation) along the three dimensions, (3) random intensity shifting and scaling, (4) coarse patch cropping, and (5) noise addition. The probability and magnitudes of each type of transformation were fine-tuned as hyper-parameters (S2 Table). All augmentations were implemented using the MONAI package, version 1.0 [32].

To perform a final assessment for the models, we froze parameter values at the best-generalised epochs and evaluated on the 30% held-out test set. To correct for the improper Loss function used in $M_{img}$ and $M_{fused}$ (Eq 2), we recalibrated the model by fitting an intercept-only logistic regression on the observed class and the predicted scores as the offset. We compared between different models the AUC and scaled Brier score (also called explained variation or index of predictive accuracy - IPA, Eq 4) and balanced accuracy (BA, Eq 3). Informal non-parametric Friedman Rank Sum tests and pairwise Durbin-Conover tests between model performances were also reported. Where possible, p-values were adjusted for multiplicity using Holm's method [33]. The Scaled Brier score is a validation score to measure the accuracy of probabilistic predictions [34]. In brief,

$$\begin{aligned} \text{BA} &= \frac{1}{2} \left( sensitivity + specificity \right) \\ &= \frac{1}{2} \left( \frac{TP}{TP + FN} + \frac{TN}{TN + FP} \right) \end{aligned} \tag{3}$$

$$\mathbf{B} = \frac{1}{N} \sum_{i=1}^{N} \left( \hat{y}_i - y_i \right)^2$$

$$\text{IPA} = \frac{\mathbf{B}_{null} - \mathbf{B}}{\mathbf{B}_{null}} = 1 - \frac{\mathbf{B}}{\mathbf{B}_{null}} \tag{4}$$

where B is the Brier score, $y_i = 0, 1$ is the observed binary outcome, $0 \leq \hat{y}_i \leq 1$ is the predicted risk $P(y_i = 1)$ given model $\hat{M}_{-i}$. $\mathbf{B}_{null}$ is the Brier score of the null model, i.e., the model that does not include any covariables (an intercept-only model). An IPA = 0 means that the model's performance is identical to that of the Null model (no benefit gained). The best value is 1 (the model has perfect prediction). An IPA < 0 happens when the use of the model is even more harmful than a random guess.

The classification threshold to quantify BA was chosen based on the apparent prevalence of the primary endpoint [35]. We reported the mean and variability of these performance metrics across models fit on 10 folds, in which:

- Validation-based metrics were measured on the corresponding left-out validation sets (i.e., 1/5 of the training parts).
- Test-based metrics were measured on the unseen test dataset (i.e., 30% of the whole sample).

Finally, based on this selected model configuration, we performed an exploratory analysis on voxel-wise contributions by using Smooth Gradient-based Saliency maps (Smooth-Grad, [36]). The procedure, implemented in MONAI, combines guided back-propagation technique with Gaussian noise smoothing to produce a heatmap of feature importance, on the voxel level. This enabled us to explore which regions in the brain were most focused on by the model and thus provided us with some suggestive points for future research.

## Results

### Baseline characteristics

The baseline characteristics of the population are summarised in Table 2. Amongst 264 participants admitted to HTD from two studies and had baseline information collected, 215 individuals were selected into the analysis. 49 participants were excluded from the analysis due to missing or unrecoverable artefacts in baseline MRIs. In the remaining population, 56 individuals were held out for testing (Fig 3).

The observed prevalence of new neurological events, including death, was 22.3%. Median age was 32 years with a 1st-3rd inter-quartile range of (24, 42) years. About half of the curated cohorts were in a mild stage, with 46% participants in MRC Grade I at enrolment and Median GCS = 15.

Out of 215 analysed participants, 79 individuals (36.7%) were infected with HIV. There were slightly more patients with MRC grade III in the HIV-positive group (6.3% than in the HIV-negative cohort (3.7%). However the difference was not statistically significant (p-value = 0.6).

Compared with HIV-negative participants, there was a considerably higher occurrence of event among those with HIV (35.4% and 14.7%, respectively, p-value < 0.001), in which a lower age and CSF lymphocyte count were also observed. There were no other apparent differences between the two HIV cohorts in our dataset.

At the time of the analysis, among 215 participants, two had missing Glasgow Coma Score (0.9%), three had missing local neurological deficit (1.4%), and eight had missing CSF lymphocyte count (3.7%) (Table 2). As outlined in Section Materials and Method, these missing data were imputed per iteration using PMM method as part of the training process [25].

Table 3 outlined the distributions of the outcome and non-imaging covariables between our training (N=159) and test (N=56) set. Overall, there was no considerable difference between the two cohorts (unadjusted p > 0.1 for all comparisons).

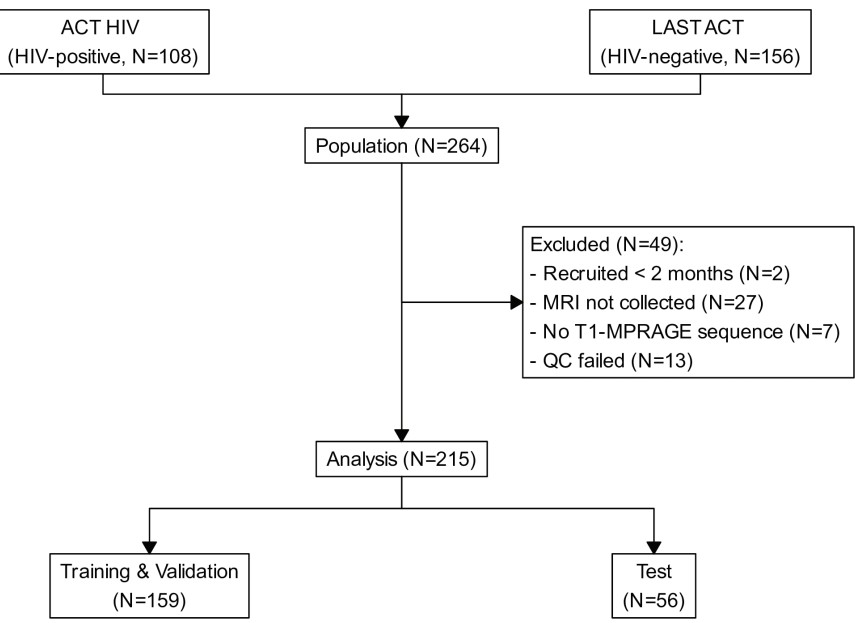

**Fig 3. Patient recruitment profile.**

**Table 3. Baseline characteristics, by data splitting to 70% training/validation and 30% testing.**

| Characteristic | N | Training N = 159 | Test N = 56 | p-value |
|---|---|---|---|---|
| Event | 215 | 37 (23%) | 11 (20%) | 0.6 |
| HIV Status | 215 | | | 0.9 |
| HIV-negative | | 100 (63%) | 36 (64%) | |
| HIV-positive | | 59 (37%) | 20 (36%) | |
| Age (years) | 215 | | | 0.9 |
| Weight (kg) | 215 | 52 (47, 60) | 50 (46, 55) | 0.4 |
| Local neurological deficit | 212 | 19 (12%) | 10 (19%) | 0.2 |
| (Missing) | | 1 | 2 | |
| GCS | 213 | 15 (14, 15) | 15 (13, 15) | 0.4 |
| (Missing) | | 0 | 2 | |
| MRC Grade | 215 | | | 0.6 |
| I | | 74 (47%) | 25 (45%) | |
| II | | 179 (50%) | 27 (48%) | |
| III | | 6 (3.8%) | 4 (7.1%) | |
| CSF lymphocyte count (cells/$\mu$L) | 207 | 77 (50, 87) | 72 (49, 86) | 0.011 |
| (Missing) | | 8 | 0 | |
| Median (1st, 3rd quartiles) for numeric variables; n (%) for categorical variables. | | | | |
| Comparision was done by Wilcoxon rank sum test for continuous variables and Pearson's Chi square test for categorical variables | | | | |
| Event is defined as any neurological event or death within 60 days of follow-up | | | | |

## Model selection and performance

Chosen hyper-parameter values for data augmentation resulting from our grid search are shown in S2 Table. In general, models which were trained under small but frequent augmentation had better generalisation than those with large but scarce augmentation. After the grid search, the optimal value for Ridge regularisation term in $M_{clin}$ was found at $\lambda = 0.5$, whereas

the optimal weight decays for $M_{img}$ and $M_{fused}$ were found at 0.12 for each feature extractor and 0.2 for the *Classifier*. In the same order, optimal dropout probabilities were 0.25 and 0.5. LR was initialised at $\lambda(0) = 4 \times 10^{-4}$ and was decayed by 5 times ($\gamma = 0.2$) every k = 400 epochs.

On the pooled validation dataset, both imaging-only and fused models showed comparable AUCs, with an average of 77% for $M_{img}$ and 78.2% for $M_{fused}$ (Table 4 and Fig 4A). Both models were apparently superior to $M_{clin}$ (AUC = 60.7%). The same trend was shown with scaled Brier score (IPA) and the BA at the threshold based on event prevalence = 22% (herein denoted as just BA), with an increase from 55.2% in $M_{clin}$ to 66.4% in the fused model.

The fold-wise comparison of AUC (Fig 4C) and Scaled Brier score (IPA, Fig 4E) showed a significant difference in performances between three models (Friedman's p-value < 0.001 for IPA and p-value = 0.01 for AUC). Post hoc pairwise comparisons showed a leap in performance from $M_{clin}$ to $M_{fused}$ (with adjusted Durbin-Conover's p-value = 0.005 for AUC and adjusted p-value < 0.0001 for IPA). However, the AUC of $M_{clin}$ widely varied across different folds with minimum AUC as low as 58% and highest AUC surpassing $M_{fused}$ with AUC > 90% (Fig 4C). The fold-based IPAs of $M_{clin}$ were more stable and consistently lower than those of $M_{fused}$ (Fig 4E).

On the test dataset (N = 56), $M_{fused}$ still performed the best on average amongst the three with AUC = 77.3% and BA = 66.4% (Table 4). $M_{img}$ had a lower AUC and IPA but higher BA compared with $M_{clin}$. Different from the pooled validation set, Table 4 also demonstrated that, across all metrics on the test set, $M_{clin}$ was the most stable with minimal variation across different folds.

The difference between the validation dataset and test dataset also holds for fold-wise comparisons (Fig 4D & F), in which Friedman's p-value < 0.001 for both IPA and AUC. $M_{clin}$ showed a superior AUC (adjusted Durbin-Conover's p-value = 0.005) and IPA (adjusted Durbin-Conover's p-value = 0.01) compared with $M_{img}$. Both models apparently had poorer discriminative value than $M_{fused}$ (adjusted Durbin-Conover's p-values < 0.001 for AUC), with some suggestive indication of a gain in IPA between $M_{clin}$ and $M_{fused}$ (adjusted Durbin-Conover's p-value = 0.24).

Table 5 shows the model evaluation on the test dataset, stratified by cohorts. In the HIV-negative cohort (N = 36), it is suggested that $M_{clin}$ had slightly lower AUC slightly higher IPA than $M_{img}$. BA of $M_{clin}$ was 55.4% ± 4.2%, apparently lower than that of $M_{img}$ (67.3% ± 2.3%). Both models were consistently inferior to $M_{fused}$ (AUC = 84.1% and BA = 35.3).

On the contrary, among participants with HIV (N = 20), $M_{clin}$ had better accuracy of 59.2% ± 5.7% than our fused model (53.3% ± 2.8%). However, the former still had marginally lower IPA and lower AUC than $M_{fused}$. $M_{img}$ apparently performed worst in this cohort, with AUC = 57% and BA = 66%.

**Table 4. Performances of the non-imaging, imaging-only, and fused model after fine-tuning, on the validation and test set. For each type of model, the performance metrics (AUC and IPA) and their variation were estimated based on** *n* = 10 **models optimized on** *n* = 10 **folds.**

| Model | Validation set | | | Test set | | |
|---|---|---|---|---|---|---|
| (%) | AUC | IPA | BA | AUC | IPA | BA |
| $M_{clin}$ | 60.7 ± 10.6 | 22.3 ± 17.4 | 58.1 ± 7.0 | 71.2 ± 1.1 | 34.2 ± 0.5 | 62.5 ± 2.1 |
| $M_{img}$ | 77.0 ± 5.4 | 29.1 ± 9.0 | 70.3 ± 7.9 | 67.3 ± 2.6 | 31.1 ± 3.6 | 61.7 ± 6.2 |
| $M_{fused}$ | 78.2 ± 6.3 | 31.5 ± 8.4 | 70.5 ± 6.0 | 77.3 ± 4.0 | 37.3 ± 2.9 | 68.6 ± 6.4 |
| Values are in the format "Mean (sd)" across 10 models. | | | | | | |
| Balanced Accuracy and IPA were calculated at 22% (event prevalence) | | | | | | |
| AUC: area under the ROC curve. IPA: index of predictive accuracy. BA: balanced accuracy | | | | | | |

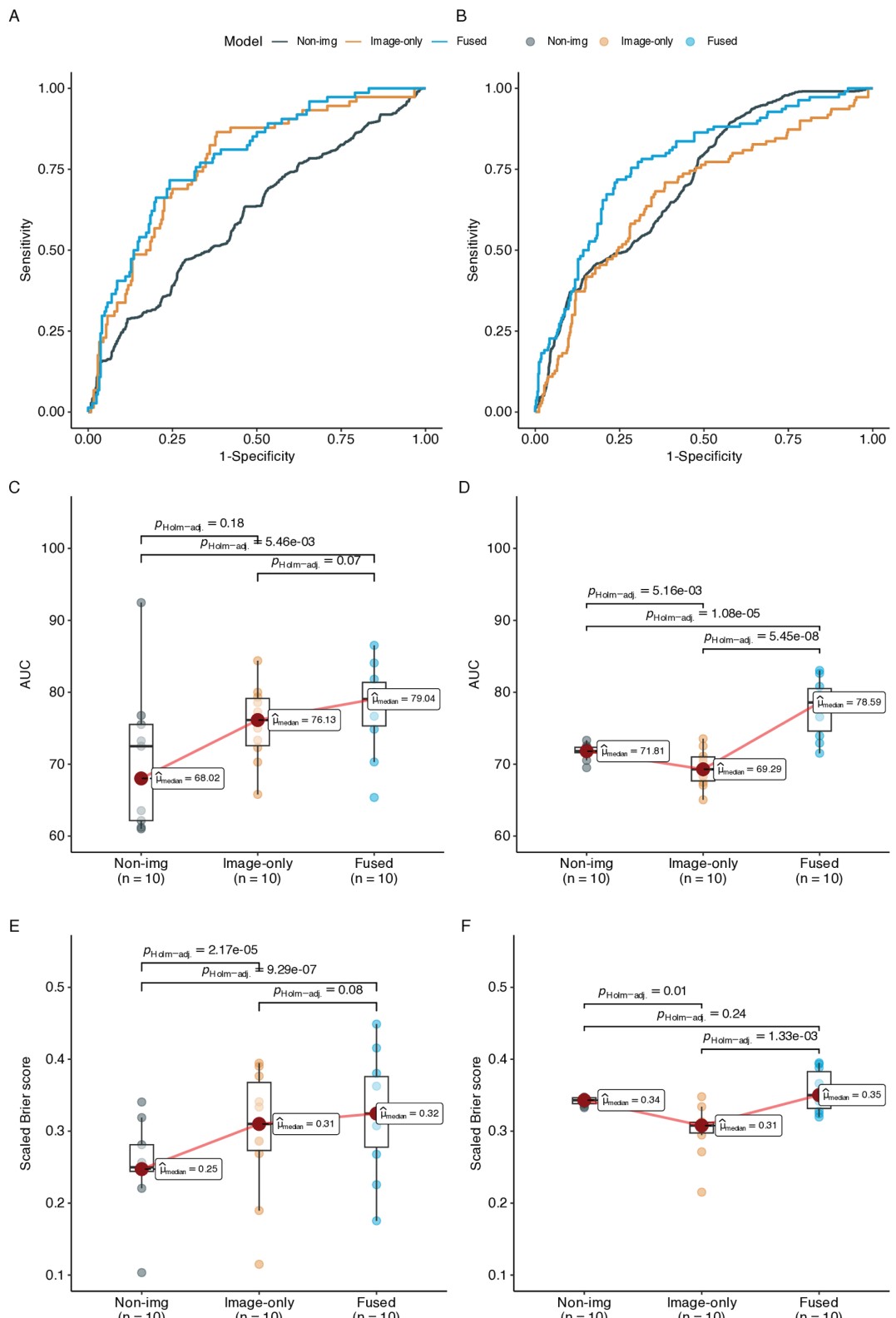

**Fig 4. Performance of our model against validation and test datasets.** Each model was trained n=10 times on 10 cross-validated folds of the 2 repeated 5-fold cross-validations. A, B: Average ROC curves on validation and test datasets, stratified by models. C, D: Box plots of AUCs on validation and test datasets across folds, stratified by models. E, F: Box plots of scaled Brier scores on validation and test datasets across folds, stratified by models.

**Table 5. Performances of the non-imaging, imaging-only, and fused model on test set (N=56), stratified by HIV status. For each type of model, the performance metrics (AUC and IPA) and their variation were estimated based on n=10 models optimised on n=10 folds.**

| Model | HIV-positive (p=35%) | | | HIV-negative (p=15%) | | |
|---|---|---|---|---|---|---|
| (%) | AUC | IPA | BA | AUC | IPA | BA |
| $M_{clin}$ | 64.3 ± 13.7 | 8.3 ± 4.2 | 59.2 ± 5.7 | 71.8 ± 10.7 | 48.5 ± 5.5 | 55.4 ± 4.2 |
| $M_{img}$ | 57.0 ± 6.1 | 5.0 ± 4.2 | 52.7 ± 8.9 | 74.4 ± 3.5 | 45.6 ± 4.1 | 67.3 ± 2.3 |
| $M_{fused}$ | 66.5 ± 5.5 | 11.7 ± 2.8 | 53.3 ± 2.8 | 84.1 ± 6.4 | 51.5 ± 2.5 | 77.3 ± 6.1 |
| Values are in the format "Mean (sd)" across 10 models. | | | | | | |
| Balanced Accuracy and IPA were calculated at 22% (event prevalence) | | | | | | |
| AUC: area under the ROC curve. IPA: index of predictive accuracy. BA: balanced accuracy | | | | | | |

The AUC of all three models were higher in HIV-negative cohorts, with AUC = 71.8% for $M_{clin}$, marginally lower than $M_{img}$ (74.4% ) and considerably lower than $M_{fused}$ (84.1%). The same trend was observed for IPA, where three models were considerably more predictive in this cohort (with IPA ≈ 45-50%) compared to the HIV-positive group (with IPA ≈ 5-11%). The only exception was that BA of $M_{clin}$ amongst HIV-negative patients was lower than the other (55.4% vs. 59.2%).

## Inferences on relevant features of the selected model

The saliency maps produced by SmoothGrad [37] demonstrate which voxels contribute the most to the decision of the $M_{fused}$ via a 3D heatmap. Each sub-figure (Fig 5) shows a 2D slice of MRI scans included in the training set, overlaid by the colour representing the level of importance of the regions to the decision, from blue to red. In general, the post hoc analysis showed a strong focus around the inner area of the cerebrum (Fig 5A, 5C), arguably the white matter. Also found to be important to the model's decision was the grey matter area around the Sylvian fissures, which overlapped with the frontmost region of the temporal lobe (Fig 5D, green arrows), the lateral and third ventricles (Fig 5A, 5B, 5D). Sub-figure A and B also highlighted some peri-ventricular brain regions, in particular the corpus callosum, the caudate nucleus, and the midbrain area (green arrows). An indistinguishable part in the cerebellum and the pons-medulla combination made a strong but not consistent contribution (Fig 5A, 5B, purple arrows).

In contrast, the existence of tuberculomas and enhancement of the basal meninges only provided a weak contribution to the model's decision and appeared as dark blue regions on the heatmap (Fig 5B, 5C, orange arrows). A stronger but still weakly highlighted tuberculoma was marked on Fig 5A (orange arrow). In general, however, the boundaries between strong and weak signal regions were unclear and not specific to any particular anatomical region of the brain.

## Discussion

With high mortality rate, forecasting TBM progress has been considered a question of utmost clinical importance. However, despite decades-long research, this still a huge clinical challenge. The search for potential prognostic biomarkers have been continuing, with high cost, but minimal success. Contradictorily, the use of magnetic resonance images (MRIs) in outcome prediction is very limited, mainly due to the lack of an efficient method to fully handle such a complicated type of biomarker.

Our study aimed to investigate this modality with data-driven techniques. Leveraging convolutional neural network, we has shown evidence for the added prognostic value of MRIs

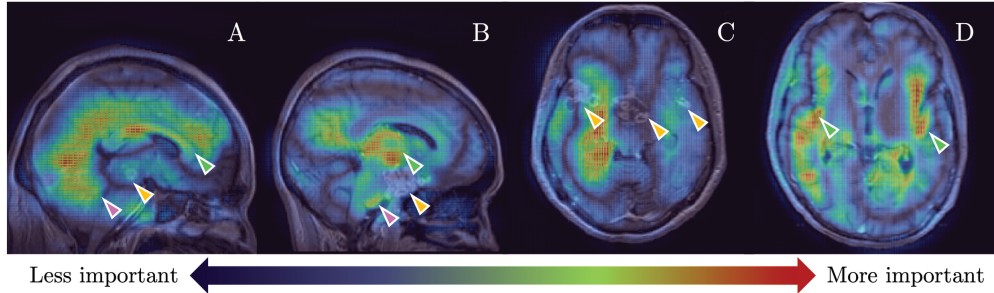

Less important ⬅ More important

**Fig 5. Sample saliency maps showing several regions guiding the model, on sagittal and transversal planes.** The redder the pixel, the more it contributed to the prediction. The model focused on the corpus callosum (A, green arrow), cerebellum (A, purple arrow), brain stem (B, green and purple arrows), and temporal lobe around Sylvian fisure (D, green arrows). The enhanced basal meninges and tuberculomas (orange arrows) did not provide a strong contribution to the decision.

to predict the poor outcome of TBM, independently and in conjuction with standard-of-care prognostic features. We developed and compared three models: one with only clinical information ($M_{clin}$), one with only MRI-extracted information ($M_{img}$), and the other where both information was fused ($M_{fused}$). The models performed well on unseen data, demonstrating their clinical value and potential.

Our findings have shown evidence that MRI alone can predict poor TBM outcome on parity with conventional biomarkers (with balanced accuracy = 61.7% for $M_{img}$ and 62.5% for $M_{clin}$). This strongly suggested that there were signals associated with TBM outcomes in both data types. Their predictive signals did not fully overlap, as when combining two modalities, $M_{fused}$ effectively gained 6.1% in both balanced accuracy and AUC, and 3.1% in IPA.

Stratified by HIV status, it is also suggestive that, in the HIV-negative group, imaging information alone could provide marginally better performance than the non-imaging counterpart, effectively gaining 2.6% in AUC and 11.9% in BA. Combining both modalities increased 12.3% in AUC and 21.9% in BA. In general, all models performed considerably better in the HIV-negative cohort. This is potentially because of less variance in the progression of TBM among the HIV-negative cohort.

Conversely, amongst patients coinfected with HIV, conventional biomarkers may play a more decisive role in predicting patient outcome, and the inclusion of imaging data did not contribute much to the prediction AUC (Table 5). One explanation could be that mortality in HIV-positive patients was more prominently driven by underlying comorbidities than just tuberculous complications on the brain alone. A notable point is that although the accuracy of $M_{fused}$ is marginally lower than $M_{clin}$ in this cohort, the AUC and IPA of the fused model were still higher than those of the non-imaging one. This can be interpreted that, although there might not be an accuracy gain in this cohort when the fused model made the right decision, it was made with higher confidence. Regardless, the limited test sample size for the HIV-positive cohort (N = 20) could also be a factor hindering any conclusive comparison. Furthermore, these results are purely descriptive without rigid statistical testing and, therefore, should be considered informal. However, further investigation could open a new insight and opportunity for TBM research.

The diagnostic and prognostic value of biomarkers based on MRIs have been studied in TBM for a long time [12,38], but with limited success. Neural networks, albeit powerful, by their black-box nature provide little understanding of the association between predictors and outcomes, which may hamper the confidence in practice and the transfer of knowledge.

In this study, SmoothGrad algorithm allowed us to look into the decision-making process of the model, with minimal assumptions. Leveraging this technique, our study found some brain areas relevant in predicting TBM outcomes, on voxel level, and ranked them by importance. Some findings including the contribution of hydrocephalus are consistent with prior findings in the literature, thus increasing our confidence in their validity.

There are, however, some contradictory areas. In particular, despite being clinically believed to be a sign of poor prognosis, the existence of tuberculomas and the enhancement of the basal meninges were not associated with bad outcomes. This can be due to the scarcity and the variability of these lesions, especially tuberculomas. Clinically, the prognostic impact of tuberculomas can vary depend on their position. For instance, their "mass effect" - a medical phenomenon caused by an alien structure pushing and displacing its surrounding tissue - can be harmless when appearing on the white matter region, but can cause an autonomous system disruption when appearing in the important body such as the brain stem or hypothalamus, as detected in Fig 5A, orange arrow. Under the limited amount of samples, our model might have failed to capture this position-dependent association. This essentially leaves room for improvement for future research.

An intriguing finding is the focus on the Sylvian fissures and the frontmost parts of the temporal lobe, the midbrain, the medulla oblongata, and the corpus callosum. These areas are important brain bodies that maintain the vital functions of the body. The temporal lobe provides a wide range of perceptive functions, including emotion, sensing, hearing, and memory generation, while the corpus callosum is believed to be bridging the two brain sides. The midbrain, the pons, and the medulla oblongata are parts of the brainstem, which coordinate various aspects of sensory and motor processing. The midbrain controls visual and auditory reflexes and is integral for motor coordination, linking the cerebellum and cerebrum. The pons acts as a relay station between the cerebrum and cerebellum, playing a key role in motor control, balance, eye movement and facial expressions and is essential for respiratory rhythm. The medulla oblongata is vital for autonomic functions, especially the cardiovascular and respiratory systems.

Any impact on these particular structures, as a consequence, has the potential to reduce cognition - which is associated with GCS, MRC Grade, and the neurological complication as defined in the primary endpoints - and autonomous dysfunction - including diabetes insipidus, hyponatremia, or the syndrome of inappropriate anti-diuretic hormone secretion (SIADH). All of these syndromes are believed to be severe complications of TBM and substantially contribute to the risk of death. These findings are consistent with the prior study where plasma sodium levels have been short-listed as an important risk factor of 9-month mortality [1] and with the pathological beliefs around TBM. By exploring these visual patterns, we could acquire better insight into the regional targets of the disease and thus develop customized treatment strategies.

Compared to a recent survival model by Thao et al. [1], our reference non-imaging model has inferior performance (AUC = 71.2% vs. 77–78%). There are several explanations for this. The first could be due to the lack of CD4+ count as a predictor in the HIV-positive group. Another potential reason is that we did not include patients who had died prematurely before any MRI scans were acquired. These patients were likely to be severe and hence presented with clear signals. The impact of this selection also notably reduced the prevalence of the outcome of interest to 22% (15% for HIV-negative and 35% for HIV-positive patients), considerably lower than the reported prevalence (1/3 for HIV-negative and 1/2 for HIV-positive patients) in other studies [1–3]. The last potential explanation is that we ignored the effect of dexamethasone on mortality at two months. This was due to the ongoing status of the RCTs at the time of analysis. Although the most recent report from the HIV-positive cohort [39] did

not show any statistically significant difference between the two treatment arms, this unadjusted variable could have induced unwanted variance and could be considered a limitation that hampered our performance.

Other limitations were also expected. First, although the original RCTs took place in the two largest tertiary hospitals in Ho Chi Minh City, Viet Nam [18,20], only participants at one site (HTD) were acquired in our analysis. The results from this study thus did not take into account the variation across different sites, although there was not expected to be one, as implied in the analysis plan and the report of ACT HIV [19,39]. Additionally, as discussed above, the limited dataset size included in the analysis increases the risk of overfitting.

The standard deviation of performance metrics also indicated a large variability across the different validation sets and across the validation and test datasets. Interestingly, the non-imaging model varied the most in the validation sets but stabilized in the test set. This is likely induced by the limited size of the validation sets (32 subjects), which possibly was too small to represent the whole training sample. One additional factor could be the regularization term we used in this model. Better stability was witnessed in the other two models where we imposed a strong regularization to promote generalizability. Such strong regularization, however, inevitably harmed the overall performance [40] as can be observed with the performance drop between the validation and the testing sets (Fig 4).

Finally, despite the consistency with past knowledge, our interpretation on the saliency maps was limited. The translation from voxel to brain region carried some level of uncertainty and subjectivity, especially when the boundary between critical and less important areas was fuzzy. A more rigid analysis based on brain segmentation is therefore necessary to confirm these findings.

One strong point of our study lays within the uniqueness of the dataset we acquired. Our datasets came from two large RCTs conducted in Southern Viet Nam dedicated to the prognosis of TBM. Every participant was tested for HIV before enrollment. Clinical and laboratory biomarkers were collected and rigorously monitored during follow-up with minimal missing data. This considerably raised the validity, reliability and statistical power of our analysis.

Another novelty in our study is the automatic extraction of MRI biomarkers in a fully data-driven manner, without depending on manual reading from radiologists. This helps the clinicians across different sites in making an objective decision. On the technical side, this study can be considered an example of incorporating multi-modal information. Leveraging fused convolutional neural network architectures and state-of-the-art techniques, we developed an end-to-end model that automatically extracts and incorporates imaging information in combination with conventional clinical and laboratory features to predict the occurrence of neurological complications, including death, within 60 days after the first imaging session. To our knowledge, this is among the first works that utilize imaging signals in a fully automatic way to improve the prognosis prediction of TBM, complementing the work by Canas et al. [41] on predicting the long-term sequelae among TBM patients. The findings in our studies have paved the way for further investigation of TBM brain imaging.

In conclusion, our study has presented a new approach to predict poor TBM outcomes, using imaging in combination with conventional biomarkers. We have shown evidence for the complementary value of MRI in predicting the survival of TBM in conjunction with conventional biomarkers. In critical conditions like TBM that affect, accurate patient triage is highly beneficial, especially in low-middle-income countries. Our study also found sound indicative regions associated with poor outcomes, paving the way to improve the understanding of disease pathology and increase clinical confidence in the prediction. To further confirm our findings, a larger study is necessary.

## Acknowledgement

We would like to thank all patients and families participating in this study, and all staff at HTD who looked after the patients.

## List of VITAL investigators

*Oxford University Clinical Research Unit, Viet Nam*: Dong Huu Khanh Trinh*, Dang Phuong Thao, Dang Trung Kien, Doan Bui Xuan Thy, Du Hong Duc, Ronald Geskus, Ho Bich Hai, Ho Quang Chanh, Ho Van Hien, Huynh Trung Trieu, Evelyne Kestelyn, Lam Minh Yen, Le Dinh Van Khoa, Le Thanh Phuong, Le Thuy Thuy Khanh, Luu Hoai Bao Tran, Luu Phuoc An, Angela Mcbride, Nguyen Lam Vuong, Nguyen Quang Huy, Nguyen Than Ha Quyen, Nguyen Thanh Ngoc, Nguyen Thi Giang, Nguyen Thi Diem Trinh, Nguyen Thi Le Thanh, Nguyen Thi Phuong Dung, Nguyen Thi Phuong Thao, Ninh Thi Thanh Van, Pham Tieu Kieu, Phan Nguyen Quoc Khanh, Phung Khanh Lam, Phung Tran Huy Nhat, Guy Thwaites, Louise Thwaites, Tran Minh Duc, Trinh Manh Hung, Hugo Turner, Jennifer Ilo Van Nuil, Vo Tan Hoang, Vu Ngo Thanh Huyen, Sophie Yacoub,

*The Hospital for Tropical Diseases, Ho Chi Minh City, Viet Nam*: Cao Thi Tam, Duong Bich Thuy, Ha Thi Hai Duong, Ho Dang Trung Nghia, Le Buu Chau, Le Mau Toan, Le Ngoc Minh Thu, Le Thi Mai Thao, Luong Thi Hue Tai, Nguyen Hoan Phu, Nguyen Quoc Viet, Nguyen Thanh Dung, Nguyen Thanh Nguyen, Nguyen Thanh Phong, Nguyen Thi Kim Anh, Nguyen Van Hao, Nguyen Van Thanh Duoc, Pham Kieu Nguyet Oanh, Phan Thi Hong Van, Phan Tu Qui, Phan Vinh Tho, Truong Thi Phuong Thao

*University of Oxford, Oxford, UK*: Natasha Ali, David Clifton, Mike English, Jannis Hagenah, Ping Lu, Jacob McKnight, Chris Paton, Tingting Zhu

*Imperial College London, London, UK*: Pantelis Georgiou, Bernard Hernandez Perez, Kerri Hill-Cawthorne, Alison Holmes, Stefan Karolcik, Damien Ming, Nicolas Moser, Jesus Rodriguez Manzano

*King's College London, London, UK*: Liane Canas, Alberto Gomez, Hamideh Kerdegari, Andrew King, Marc Modat, Reza Razavi, Miguel Xochicale

*Ulm University, Germany*: Walter Karlen

*The University of Melbourne, Australia*: Linda Denehy, Thomas Rollinson

*Mahidol Oxford Tropical Medicine Research Unit, Thailand*: Luigi Pisani, Marcus Schultz

## List of abbreviation

| | |
|---|---|
| AUC | Area under the (ROC) curve |
| BA | Balanced accuracy |
| CNN | Convolutional neural network |
| CSF | Cerebrospinal fluid |
| FC | Fully connected (layer) |
| GCS | Glasgow coma score |
| HIV | Human immunodeficiency virus |
| HPC | High performance computing |
| HTD | Hospital for Tropical Diseases |
| IPA | Index of predictive accuracy |
| JADE | Joint Academic Data Science Endeavour |
| MNI | Montreal Neurosciences Institute |
| MRC | (British) Medical Research Council |
| MRI | Magnetic resonance image |

| | |
|---|---|
| Mtb | Mycobacterium tuberculosis |
| LR | Learning rate |
| QC | Quality check |
| RCT | Randomised control study |
| ROC | Receiver operating characteristic |
| TB | Tuberculosis |
| TBM | Tuberculous meningitis |
| VITAL | Vietnam Intensive Care Unit Translation Applications Laboratory |

## Supporting information

**S1 Fig. Hyperparameter grid search and evaluation strategy.** Each model was trained on 4 in 5 folds in the whole training set and was used to predict the one held-out fold. All predictions and ground truths on these held-out folds were subsequently concatenated to recover the full training set (herein denoted as pooled validation set) to calculate AUC and CE loss. The hyperparameter set that maximized AUC and minimized CE loss were selected.

**S1 Table. Definition of hyper-parameters that were grid-searched by cross-validations.** Values that minimized the validation loss were selected into the final model.

**S2 Table**. **Hyper-parameters for image augmentation that were searched**, with the results of the grid search reported on column "Value."

## Author contributions

**Conceptualization:** Trinh Huu Khanh Dong, Guy E. Thwaites.

**Data curation:** Trinh Huu Khanh Dong, Joseph Donovan, Daniel Beasley, Nguyen Hoan Phu, Nguyen Thi Ha.

**Formal analysis:** Trinh Huu Khanh Dong.

**Funding acquisition:** Nguyen Thuy Thuong-Thuong, Reza Razavi, Guy E. Thwaites.

**Investigation:** Joseph Donovan, Nguyen Thi Ha, Guy E. Thwaites.

**Methodology:** Trinh Huu Khanh Dong, Liane S. Canas, Marc Modat.

**Project administration:** Nguyen Thuy Thuong-Thuong, Sebastien Ourselin, Reza Razavi, Guy E. Thwaites, Marc Modat.

**Resources:** Daniel Beasley, Nguyen Thuy Thuong-Thuong.

**Software:** Trinh Huu Khanh Dong, Liane S. Canas, Daniel Beasley.

**Supervision:** Nguyen Thuy Thuong-Thuong, Reza Razavi, Guy E. Thwaites, Marc Modat.

**Validation:** Trinh Huu Khanh Dong, Guy E. Thwaites, Marc Modat.

**Visualization:** Trinh Huu Khanh Dong.

**Writing – original draft:** Trinh Huu Khanh Dong.

**Writing – review & editing:** Trinh Huu Khanh Dong, Liane S. Canas, Joseph Donovan, Nguyen Thuy Thuong-Thuong, Nguyen Hoan Phu, Sebastien Ourselin, Reza Razavi, Guy E. Thwaites, Marc Modat.

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
