## [Decision Letter · Decision Letter 0]

28 Nov 2024

PONE-D-24-44674Convolutional neural network using magnetic resonance brain imaging to predict outcome from tuberculosis meningitisPLOS ONE

Dear Dr. Dong,

Thank you for submitting your manuscript to PLOS ONE. After careful consideration, we feel that it has merit but does not fully meet PLOS ONE’s publication criteria as it currently stands. Therefore, we invite you to submit a revised version of the manuscript that addresses the points raised by two reviewers below.

A rebuttal letter that responds to each point raised by the reviewers. You should upload this letter as a separate file labeled 'Response to Reviewers'.A marked-up copy of your manuscript that highlights changes made to the original version. You should upload this as a separate file labeled 'Revised Manuscript with Track Changes'.An unmarked version of your revised paper without tracked changes. You should upload this as a separate file labeled 'Manuscript'.

We look forward to receiving your revised manuscript.

Kind regards,

Steve Graham

Stephen Michael Graham, FRACP, PhD

Academic Editor

PLOS ONE

Journal requirements: When submitting your revision, we need you to address these additional requirements. 1. Please ensure that your manuscript meets PLOS ONE's style requirements, including those for file naming. The PLOS ONE style templates can be found at https://journals.plos.org/plosone/s/file?id=wjVg/PLOSOne_formatting_sample_main_body.pdf and https://journals.plos.org/plosone/s/file?id=ba62/PLOSOne_formatting_sample_title_authors_affiliations.pdf 2. Thank you for stating the following financial disclosure:  [T.D, J.D, N.T.T.T, N.T.H, L.S.C, D.B, S.O, R.R, G.E.T, M.M: Wellcome Trust (215010/Z/18/Z, 217650/Z/19/Z, 225167/Z/22/Z), L.S.C, D.B, S.O, R.R, M.M: Wellcome/EPSRC Centre for Medical Engineering (203148/Z/16/Z) All data processing was run on the Wellcome Trust/EPSRC-supported (EP/T022205/1) Joint Academic Data Science Endeavour (JADE) HPC cluster.].  Please state what role the funders took in the study.  If the funders had no role, please state: ""The funders had no role in study design, data collection and analysis, decision to publish, or preparation of the manuscript."" If this statement is not correct you must amend it as needed. Please include this amended Role of Funder statement in your cover letter; we will change the online submission form on your behalf. 3. Thank you for stating the following in the Acknowledgments Section of your manuscript: [We would like to thank all patients and families participating in this study, as well as all staff at HTD who looked after the patients. This project received funding from the Wellcome Trust (215010/Z/18/Z), the Wellcome/EPSRC Centre for Medical Engineering (203148/Z/16/Z), the AI Centrefor Value-Based Healthcare, the Vietnam ICU Translation Applications Laboratory (VITAL) consortium (225167/Z/22/Z) and Oxford University Clinical Research Unit(217650/Z/19/Z). All processing was run on the EPSRC-supported (EP/T022205/1)Joint Academic Data Science Endeavour (JADE) HPC cluster]We note that you have provided funding information that is not currently declared in your Funding Statement. However, funding information should not appear in the Acknowledgments section or other areas of your manuscript. We will only publish funding information present in the Funding Statement section of the online submission form. Please remove any funding-related text from the manuscript and let us know how you would like to update your Funding Statement. Currently, your Funding Statement reads as follows:   [T.D, J.D, N.T.T.T, N.T.H, L.S.C, D.B, S.O, R.R, G.E.T, M.M: Wellcome Trust (215010/Z/18/Z, 217650/Z/19/Z, 225167/Z/22/Z), L.S.C, D.B, S.O, R.R, M.M: Wellcome/EPSRC Centre for Medical Engineering (203148/Z/16/Z) All data processing was run on the Wellcome Trust/EPSRC-supported (EP/T022205/1) Joint Academic Data Science Endeavour (JADE) HPC cluster.].   Please include your amended statements within your cover letter; we will change the online submission form on your behalf. 4. We note that you have indicated that there are restrictions to data sharing for this study. PLOS only allows data to be available upon request if there are legal or ethical restrictions on sharing data publicly. For more information on unacceptable data access restrictions, please see http://journals.plos.org/plosone/s/data-availability#loc-unacceptable-data-access-restrictions.  Before we proceed with your manuscript, please address the following prompts: a) If there are ethical or legal restrictions on sharing a de-identified data set, please explain them in detail (e.g., data contain potentially identifying or sensitive patient information, data are owned by a third-party organization, etc.) and who has imposed them (e.g., a Research Ethics Committee or Institutional Review Board, etc.). Please also provide contact information for a data access committee, ethics committee, or other institutional body to which data requests may be sent. b) If there are no restrictions, please upload the minimal anonymized data set necessary to replicate your study findings to a stable, public repository and provide us with the relevant URLs, DOIs, or accession numbers. For a list of recommended repositories, please seehttps://journals.plos.org/plosone/s/recommended-repositories. You also have the option of uploading the data as Supporting Information files, but we would recommend depositing data directly to a data repository if possible. We will update your Data Availability statement on your behalf to reflect the information you provide. 5. In the online submission form, you indicated that [The datasets used during the current study contain sensitive identifiers from the two clinical trials and thus are available on reasonable request in accordance with the trials data sharing statements. Data are parts of the two randomised control trials: the ACT HIV (identifier NCT03092817) and LAST ACT (identifier NCT03100786). Kindly contact Dr. Joseph Donovan <Joseph.Donovan@lshtm.ac.uk> or Prof. Guy Thwaites <gthwaites@oucru.org> for data sharing permission.]. All PLOS journals now require all data underlying the findings described in their manuscript to be freely available to other researchers, either 1. In a public repository, 2. Within the manuscript itself, or 3. Uploaded as supplementary information.This policy applies to all data except where public deposition would breach compliance with the protocol approved by your research ethics board. If your data cannot be made publicly available for ethical or legal reasons (e.g., public availability would compromise patient privacy), please explain your reasons on resubmission and your exemption request will be escalated for approval.  6. One of the noted authors is a  consortium [Vietnam ICU Translation Applications Laboratory (VITAL) consortium]. In addition to naming the author group, please list the individual authors and affiliations within this group in the acknowledgments section of your manuscript. Please also indicate clearly a lead author for this group along with a contact email address.

Reviewers' comments:

Reviewer's Responses to Questions

**Comments to the Author**

1. Is the manuscript technically sound, and do the data support the conclusions?

Reviewer #1: Yes

Reviewer #2: Yes

2. Has the statistical analysis been performed appropriately and rigorously? 

Reviewer #1: I Don't Know

Reviewer #2: Yes

3. Have the authors made all data underlying the findings in their manuscript fully available?

Reviewer #1: Yes

Reviewer #2: No

4. Is the manuscript presented in an intelligible fashion and written in standard English?

Reviewer #1: Yes

Reviewer #2: Yes

5. Review Comments to the Author

Reviewer #1: I find this manuscript highly interesting and promising, particularly in its potential to enhance the diagnostic utility of MRI as an additional tool for tuberculosis meningitis (TBM). However, I have several comments for consideration:

Accessibility for Clinicians: The manuscript is highly technical, which may limit its practical application for clinicians in routine care. Simplifying the language and providing more direct clinical insights would enhance its relevance and usability in daily practice.

Definition of Outcomes: The definition of "events" as outcomes in this paper seems somewhat arbitrary. Death and neurological deficits represent distinct outcomes with a wide range of severity. It would be beneficial for the authors to provide a clear justification for grouping these outcomes together or, alternatively, analyze them separately.

MRI Findings and Model Explanation: The discussion of the MRI findings and the defined model is insufficiently detailed. The authors should expand on how specific MRI lesions relate to clinical features and outcomes, providing more thorough explanations and discussions to strengthen the paper's conclusions.

Reviewer #2: In this manuscript, the authors used a convolutional neural network to predict outcome from tuberculosis meningitis using imaging data only or fused with clinical and laboratory data. This is an interesting study in which the authors show that MRI has additional information in predicting the occurrence of death or neurological complications in the first two months after the MRI.

However, there are a number of issues that the authors should clarify.

The study of Canas et al. (ref 40) is using the same dataset and it should be mentioned in the beginning what the difference is with this study since there is clearly some overlap. In the discussion one cannot use this study to discuss similarities since this is expected if you use similar methodology on the same dataset. This part of the discussion should be removed.

Performance values were compared between the HIV positive and HIV negative group but this is only descriptive and no statistical analysis is performed. Without a proper statistical analysis, we cannot conclude much from a descriptive observation.

Specific comments

Figure 1: The arrow in panel C is not very well visible and this can be improved.

Who checked the quality of the MRI? Was it visually checked by a neuroradiologist? How many scans were excluded? The authors mention that a patient was excluded if no other scan was available as a replacement but this is unclear. Did you included other sequences besides a T1w-MPRAGE or did some patients have multiple scans? If the latter is true, which image was used or were all images used?

Missing variables were imputed within the training process and this is a valid technique provided that the number of missing data is small compared the collected data. How many missing data were present for the different variables?

MRI scans were rigidly registered into MNI space but I assume that scaling was applied as well. This could be mentioned explicitly.

Intensities were normalized to range [0,1]? How was this done exactly? Scaling with respect to the maximum in the image can be a problem depending where this voxel is located.

It would be interesting to see a correlation matrix between all the non-imaging features. How are features which are highly correlated (if present) handled?

The caption of figure 2 is very minimal and should be extended.

It is unclear to me how the vector of 512 imaging features is compressed into a latent vector of length 24.

P7, line 206: it is mentioned that equation 2 (with alpha = beta = 1) is analogous to concurrently maximizing the likelihood of the three tasks but I am not sure if this is indeed the case. I would expect that this depends on the relative loss with respect to the others.

P7, line 232: the model was recalibrated by fitting an intercept-only logistic regression on the observed class and the predicted scores as the offset in order to correct for class imbalance. Is there a reference for this procedure showing that this will indeed correct for class imbalance?

It is not clear what the smooth gradient based saliency maps are representing? Is it some sort of relative weight of a voxel which reflects its contribution to the final output?

In table 3, event should be replaced by new neurological event (including death) within 2 months or is it representing something else? The age of 24 (1st quartile for the whole group) is different form the value in the text which is 25. I would also add a separate table for the test set to see how representative this test set is.

P10, line 271 it is mentioned that 79 out of 215 participants (50.2%) were infected with HIV but I don’t understand what the 50.2% is representing. 79 out of 215 is much smaller than 50%.

Is there a reference or an argument why the authors selected some of the hyperparameters such as weight decay at 0.12 for each feature?

Figure 5: the quality of the figure is poor and values and text were difficult to read. I would also recommend to use the same scale for panels C and D and for panels E and F so that the reader can more easily compare the results of the validation and test set. The n=10 is confusing. I would rather mention the actual number of patients used.

In figure 6, I do not see coronal planes while this is mentioned in the caption. I would also add a colour bar.

6. PLOS authors have the option to publish the peer review history of their article (what does this mean?). If published, this will include your full peer review and any attached files.

Reviewer #1: **Yes: **Sofiati Dian., M.D., PhD

Reviewer #2: No

---

## [Author Response · Author response to Decision Letter 1]

7 Jan 2025

Overall: We would love to express our appreciation to both reviewers for your kind and detailed comments to improve the quality of our manuscript. We acknowledge that there are some issues with the content, as pointed out by the reviewers. We, hence, have carefully taken care of your points of correction and made replies and corrections on our manuscripts, where appropriate. Please kindly find below our point-by-point responses and the related lines in the resubmitted manuscript with tracked changes, as requested by the journal. To produce the tracked change version, we used package `latexdiff` as recommended. Removed text is marked with red colour and strike-through, while added text is marked in blue. Due to the complexity of the table structure, changes made within table cells were not marked automatically. We have highlighted in blue the changes manually.

We have made point-by-point responses to each reviewer, in a dedicated PDF file, and in text below, where "Q" represents the comment form the Reviewers and "R" represents our Responses. We hope that out responses will be satisfying to the Reviewers. Our deepest apologies to the reviewers for any inconvenience with the aesthetics of the change-tracked file. Please kindly refer to the untracked LaTeX/PDF version for the truly intended visual appearance and structure. I also provide a PDF version of the response letter, in which I noted down the lines where the changes were made.

# Reviewer 1

- Q: Accessibility for Clinicians: The manuscript is highly technical, which may limit its practical application for clinicians in routine care. Simplifying the language and providing more direct clinical insights would enhance its relevance and usability in daily practice.

- R: Thank you for your considerate comments. As a clinician by training, I admit that the technical level of this manuscript is quite high, which hampers the accessibility for the clinicals. During the time of writing, we have attempted to balance the technical transparency and the medical impact. However, there are some technical aspects that we cannot sacrifice for the transparency of the methodologies, which also an aspect to determine the credibility of the results.

However, we agree with the Reviewer that, we need to add some additional medical input to target the clinicians in the field. We have hence added some medical points in the Discussion part, which also associates with your third points.

- Q: Definition of Outcomes: The definition of "events" as outcomes in this paper seems somewhat arbitrary. Death and neurological deficits represent distinct outcomes with a wide range of severity. It would be beneficial for the authors to provide a clear justification for grouping these outcomes together or, alternatively, analyze them separately.

- R: Thank you for your concern. The definition of the events in general followed the definition of primary endpoints in the two main RCTs that provided the dataset (ACT HIV and LAST ACT) [20, 21]. In particular, the protocol of LAST ACT stated that “The primary endpoint is death or new neurological event (defined as a fall in Glasgow coma score (GCS) by ≥2 points for ≥2 days from the highest previously recorded GCS (including baseline) or the onset of any of the following clinical adverse events: cerebellar symptoms, focal neurological signs, or new seizures) during 12 months from randomisation.”

From a clinical perspective, as neurological complications ignited by TBM are, more likely than not, irreversible. Survivors with neurological events during treatment course tend to suffer long-term cognitive impairment and limit in their activities afterwards. The combination of both outcomes highlights that they are both clinically meaningful as a TBM outcome.

From a modelling perspective, separating the two outcomes would complicate the model, as the two outcomes can be overlapped (those who had neurological events could die consequently), diluting the main objective without much gain in the clinical impact.

- Q: MRI Findings and Model Explanation: The discussion of the MRI findings and the defined model is insufficiently detailed. The authors should expand on how specific MRI lesions relate to clinical features and outcomes, providing more thorough explanations and discussions to strengthen the paper's conclusions.

- R: We thank the Reviewer for pointing this out. As the MRI finding, in particular the saliency maps, are purely post hoc analysis without uncertainty estimation, we were extremely careful not to over-interpret the results. We acknowledge that it may be more accessible to the clinician with more clinical input. We therefore added some details regarding our observations from the post hoc analysis and provided some additional details on the corresponding figures.

However, we also highlight that, these findings are purely observation and not rigorously checked. To overcome this limitation, we also had another paper in plan which leverages brain parcellation to fully segmentate the brain regions. By having a clear definition of brain regions, we can make the interpretation more credible and accessible. This uses a completely different method and will come in a later paper.

# Reviewer 2:

- Q: The study of Canas et al. (ref 40) is using the same dataset and it should be mentioned in the beginning what the difference is with this study since there is clearly some overlap. In the discussion one cannot use this study to discuss similarities since this is expected if you use similar methodology on the same dataset. This part of the discussion should be removed.

- R: I would like to express my thank for the Reviewer’s comment. Both manuscripts were two parts of a three-part analysis where we performed different investigation on imaging biomarkers for TBM. The datasets indeed shared some overlap, but the methodologies and the outcome were different. In this manuscript, we investigated the short-term outcome during treatment course (6 week) of dexamethasone using only baseline MRI while Liane et.al. used time series MRI to predict long-term quality of life after discharge. Initially, we thought it was of interest to share some similarity in findings using different methods. However, I admitted that the Reviewer’s concern is very sensible and understandable. We hence decided to remove the related paragraph from the Discussion.

- Q: Performance values were compared between the HIV positive and HIV negative group but this is only descriptive and no statistical analysis is performed. Without a proper statistical analysis, we cannot conclude much from a descriptive observation.

- R: As the sample for the subgroup analysis was small, we constrained ourselves from performing too many statistical testing, because (1) multiplicity correction may bear the risk of nullifying our findings and (2) we did not how to perform correction in this particular case, as the tests were not independent. A potential method is to “pretend” that this was a sequential setup (HIV+ and then HIV-) and performed a Spiessens-Debois correction (https://pubmed.ncbi.nlm.nih.gov/20832503/) which exploits the correlation between them to save some error type I. However, this is (1) complicated and (2) not guaranteed to be fully correct.

We agreed with the Reviewer. Your concern is absolutely valid. We hence therefore added to the Discussion, stating that this comparison is purely speculative.

- Q: Figure 1: The arrow in panel C is not very well visible and this can be improved.

- R: Thank you for your comment. We have made the arrow bigger. Please also kindly note that I have noticed that the quality of the figures sent by the journal is not ideal. The original figures had at least 300 dpi – usually 600 dpi.

- Q: Who checked the quality of the MRI? Was it visually checked by a neuroradiologist?

- R: We (the authors) performed the visual check on the correctness of sequence, artefact (namely Gibbs noise, contrast, bias field, or any irrecoverable deformation) and field of view of the scans (missing slices, invasive crop). We did not check for any lesion appearance as it might bias the analysis. We have TBM clinicians and medical image engineers in the team who had experience with MRI.

- Q: How many scans were excluded?

- R: The total number of excluded sessions were 49, amongst which 13 were due to QC failure. The numbers were explicitly mentioned in Fig 4 and on line 275.

Scan-wise, there were in total 56 T1W scans excluded from the analysis (36 HIV-negative, 20 HIV-positive). In most of the cases, the radiographers would have noticed that the scan were unusable and performed an alternative scan in the same session, explaining the fact that there were more scans excluded than sessions. As there were usually alternative usable scans, we believe that the number of excluded scans was not relevant to the analysis.

- Q: The authors mention that a patient was excluded if no other scan was available as a replacement but this is unclear. Did you included other sequences besides a T1w-MPRAGE or did some patients have multiple scans? If the latter is true, which image was used or were all images used?

- R: My apologies we did not make this point clear. We only curated T1w-MPRAGE sequence. If the participants had multiple T1w-MPRAGE sequence, we got the first session, not later than one month (as advised by the TBM experts) nearest to the enrolment date. If there were many T1w-MPRAGE scans in the first session, we got the scan with best quality by manual QC.

Any T1w-MPRAGE scan that did not pass the manual QC would be excluded. If all T1w-MPRAGE was removed (due to this reason), the patient was excluded.

If the patient died before any MRI was taken, they weren’t included in the analysis.

- Q: Missing variables were imputed within the training process and this is a valid technique provided that the number of missing data is small compared the collected data. How many missing data were present for the different variables?

- R: In Table 3 (Baseline Characteristics), column N represented the number of observed data points for that particular variable. An N < 215 would indicate missing data. We apologies that was an implicit representation and we apology for any confusion this may have made. We hence explicitly showed the number of missing points in a dedicated row in Table 3, denoted as “(Missing)”, representing the number of missing data points for the variables preceding it, stratified by HIV status.

- Q: MRI scans were rigidly registered into MNI space but I assume that scaling was applied as well. This could be mentioned explicitly.

- R: We sorry for the confusion and have changed the wording to “affine-registered”.

- Q: Intensities were normalized to range [0,1]? How was this done exactly? Scaling with respect to the maximum in the image can be a problem depending where this voxel is located.

- R: Thank you. That is a great question. We believe that we need to scale the MRIs into some comparable intensity scale. However, choosing which is still a question under debate. We opted in a trial-error strategy by experimenting with both scaling by range and by standard deviation (sd) before feeding into the neural network.

For some unknown reason, we found that the model training was more likely to fail with sd-scaled images (stuck and did not move) When it did not fail, we found effectively no difference in performance compared with the range-scaled images. To ensure that the transformation did not corrupt the image, we also visually checked the scaled and registered images for validity before inputting them into the model. We have clarified that in the text.

In conclusion, we performed the following scaling, where x(i) is the intensity at voxel i: T(x(i))=x(i)/max(x(i))

- Q: It would be interesting to see a correlation matrix between all the non-imaging features. How are features which are highly correlated (if present) handled?

- R: This is the correlation between non-imaging continuous features (age, weight, GCS, and CSF lymphocyte count). There are some weak correlations between variables, as shown below.

age weight gcs csf_lym

age 1.00 0.12 -0.10 0.09

weight 0.12 1.00 0.00 0.15

gcs -0.10 0.00 1.00 0.15

csf_lym 0.09 0.15 0.15 1.00

While multicollinearity may interfere the association between the outcome and one particular feature, we expect them to have minimal impact on predictive value of the model (as explained here https://stats.stackexchange.com/questions/361247/multicollinearity-and-predictive-performance). As we did not perform inference on the non-imaging features, we did not perform any correction for multicollinearity.

As for the inference on the imaging feature, we used SmoothGrad, which is a non-linear technique based on the gradient and not the coefficients. As the gradient was calculated altogether for all feature point (both imaging and non-imaging), any collinearity, in theory, should have been taken into account. However, we are not sure if in practice that statement is true, as it needs further research.

- Q: The caption of figure 2 is very minimal and should be extended.

- R: We thank the Reviewer for the great comment. We have extended the figure captions.

- Q: It is unclear to me how the vector of 512 imaging features is compressed into a latent vector of length 24.

- R: This is a layer of a feed-forward neural network, effective a matrix multiplication between a matrix whose dimension = 24 x 512 (parameters) and a vector of length 512 (features). See also: https://en.wikipedia.org/wiki/Matrix_multiplication

We also made add a vector of 512 into Fig 2 to make this layer more clearly.

- Q: P7, line 206: it is mentioned that equation 2 (with alpha = beta = 1) is analogous to concurrently maximizing the likelihood of the three tasks but I am not sure if this is indeed the case. I would expect that this depends on the relative loss with respect to the others.

- R: That is a valid point. We are sorry for the confusion.

The idea is, as Cross Entropy loss is effective the Negative Log Likelihood of a binomial regression (https://en.wikipedia.org/wiki/Cross-entropy#Relation_to_maximum_likelihood), summing up the Cross Entropy loss would, in theory, summing up the Negative Log Likelihood.

As log(x)+log(y) = log(xy), we effective have the log of multiplication of the likelihood.

Minimising the total Cross Entropy loss would maximise the muliplication of likelihood. If the three tasks are independent, this is translatable to maximising the likelihood function of each tasks, as

P(A) x P(B) = P(AB) given A&B are iid.

However, we noticed that the three tasks are not mutual independent (only M_img and M_clin are independent), the formula did not translate well to our statement. Therefore, we decided to remove the confusing statement.

The formulation/treatment is still useful to constrain the model to balancely optimise the two auxilary tasks Lclin and Limg while still make them contribute to the other (by optimising the Lfused).

However, this formulation comes with a trade-off: the loss of calibration to the large, which is a characteristics of binomial regression when likelihood is maximised (https://stats.stackexchange.com/questions/390487/why-is-logistic-regression-well-calibrated-and-how-to-ruin-its-calibration). We denote this a improper loss function. This is linked to another question of yours that we will discuss below.

- Q: P7, line 232: the model was recalibrated by fitting an intercept-only logistic regression on the observed class and the predicted scores as the offset in order to correct for class imbalance. Is there a reference for this procedure showing that this will indeed correct for class imbalance?

- R: Thank you very much for pointing this out, this is a typing mistake from us. It should be “to correct for improper loss function”. We have made the change accordingly.

As discussed above, combining the losses makes the model miscalibrated due to a bias loss. This may hold true (but not as severe) with Ridge loss in penalised logistic regression. The level of miscalibration depends on the penalty term value, whic

---

## [Decision Letter · Decision Letter 1]

28 Jan 2025

PONE-D-24-44674R1Convolutional neural network using magnetic resonance brain imaging to predict outcome from tuberculosis meningitisPLOS ONE

Dear Dr. Dong.

Apologies for delay but was awaiting comments from a second reviewer who previously reviewed. Given time lapse, I have decided to proceed with comments of the one reviewer who has again carefully reviewed your manuscript. Thank you for submitting your manuscript to PLOS ONE. After careful consideration, we feel that it has merit but does not fully meet PLOS ONE’s publication criteria as it currently stands. Therefore, we invite you to submit a revised version of the manuscript that addresses the points raised during the review process.

We look forward to receiving your revised manuscript.

Kind regards,

Steve

Stephen Michael Graham, FRACP, PhD

Academic Editor

PLOS ONE

Journal Requirements:

Reviewers' comments:

Reviewer's Responses to Questions

**Comments to the Author**

1. If the authors have adequately addressed your comments raised in a previous round of review and you feel that this manuscript is now acceptable for publication, you may indicate that here to bypass the “Comments to the Author” section, enter your conflict of interest statement in the “Confidential to Editor” section, and submit your "Accept" recommendation.

Reviewer #1: All comments have been addressed

2. Is the manuscript technically sound, and do the data support the conclusions?

Reviewer #1: Yes

3. Has the statistical analysis been performed appropriately and rigorously? 

Reviewer #1: I Don't Know

4. Have the authors made all data underlying the findings in their manuscript fully available?

Reviewer #1: No

5. Is the manuscript presented in an intelligible fashion and written in standard English?

Reviewer #1: Yes

6. Review Comments to the Author

Reviewer #1: First, the author has tried to address all the reviewer's comments and requests; however, the absence of a tracked changes file makes it more challenging to follow the revisions.

Secondly, the overall clinical applicability of the article remains limited, potentially making it more relevant to a statistical audience than to clinicians. The extensive focus on methodology may diminish its practical value for clinical implementation. I recommend that the author revise the article's objective, replacing phrases such as "predict the occurrence of any neurological complication" or "expand the insight in clinical prognostications" with a more technically precise and focused expression. Alternatively, the author could enhance the article by including clear clinical conclusions in the first paragraph of the discussion and elaborating on the clinical implications in the conclusion section at the end of the discussion.

7. PLOS authors have the option to publish the peer review history of their article (what does this mean?). If published, this will include your full peer review and any attached files.

Reviewer #1: **Yes: **Sofiati Dian

---

## [Author Response · Author response to Decision Letter 2]

9 Mar 2025

Dear Reviewer,

We appreciate the Reviewer’s comment and agree that the previous version of the manuscript placed a significant emphasis on the method, mainly for transparency and reproducibility purposes. In summary, the Reviewer made two points: the lack of a tracked-changes version (point 1) and the overly focus on methodology (point 2). We will first focus on point 2, which is the primary concern of the Reviewer.

For point 2, we agree that this overfocus on methodology might have given the manuscript a less attractive position to clinical readers. Please kindly note that while our proposed approach and full framework are novel in their application to the present clinical problem of tuberculous meningitis, the core model itself is not entirely new and has been applied to other diseases. To address that point, and by aiming to refocus the paper on its clinical contribution, we have moved details regarding the method to supplementary material, from both the Methods and Results sections as follows:

- Table 2 moves to the Supplementary document and becomes Supplementary Table S1.

- Table 5 moves to the Supplementary document and becomes Supplementary Table S2.

- Figure 3 moves to the Supplementary document and becomes Supplementary Figure S1.

Furthermore, to emphasise the clinical value of our findings, we have modified the Discussion to describe better the clinical conclusions aroused from our results, as kindly suggested by the Reviewer, as follows:

- In the first paragraph, we have reiterated the clinical context and challenges that motivated our study.

- In the second paragraph, we have restated our clinical objective and introduced the main clinical results for the objective.

- In the third to eighth paragraphs, we discussed the results in detail.

- In the ninth to fourteenth paragraphs, we discussed our pros and cons.

- In the last paragraph, we reiterated our study and clinical impact.

We hope this addresses the Reviewer’s main concerns and makes our study message clearer to the journal audience.

For point 1 made by Reviewer regarding the absence of a tracked-change file, please kindly accept our deepest apologies. We have contacted the journal's editor. However, they did not find any issues. Please kindly find a PDF document with the name “Revised Manuscript with Track Changes.pdf”. If you are receiving a long PDF file with all the forms, manuscript, and download links to supplementary documents embedded, please kindly scroll down to the near bottom. I would assume you can find the tracked-changes version there, followed by a PDF version of this Response letter. For some strange reason, I am not allowed to reorder the documents when I uploaded. In the case it does not appear on your side, please kindly contact the editor or the journal, since the existence of this file is mandated by them during the resubmission process. For the resolution of the figures, please kindly find in the attached file Figure.zip where Figures are sent in their original resolution. The downscaling of the figures was not intentional. Please kindly refer to the untracked LaTeX/PDF version for the true intended visual appearance and structure.

With best regards,

---

## [Editor Report · Decision Letter 2]

10 Mar 2025

Convolutional neural network using magnetic resonance brain imaging to predict outcome from tuberculosis meningitis

PONE-D-24-44674R2

Dear Dr. Trinh Dong,

We’re pleased to inform you that your manuscript has been judged scientifically suitable for publication and will be formally accepted for publication once it meets all outstanding technical requirements.

Kind regards,

Steve

Stephen Michael Graham, FRACP, PhD

Academic Editor

PLOS ONE
---

## [Editor Report · Acceptance letter]

PONE-D-24-44674R2

PLOS ONE

Dear Dr. Dong,

I'm pleased to inform you that your manuscript has been deemed suitable for publication in PLOS ONE. Congratulations! Your manuscript is now being handed over to our production team.

Kind regards,

on behalf of

Dr. Stephen Michael Graham

Academic Editor

PLOS ONE